# Management of the Adverse Effects of Immune Checkpoint Inhibitors

**DOI:** 10.3390/vaccines8040575

**Published:** 2020-10-01

**Authors:** Manuel Morgado, Ana Plácido, Sandra Morgado, Fátima Roque

**Affiliations:** 1Pharmaceutical Services of University Hospital Center of Cova da Beira, 6200-251 Covilhã, Portugal; sandracristinamorgado@gmail.com; 2Research Unit for Inland Development, Polytechnic of Guarda (UDI-IPG), 6300-559 Guarda, Portugal; anaplacido@ipg.pt (A.P.); froque@ipg.pt (F.R.); 3Health Sciences Research Centre, University of Beira Interior (CICS-UBI), 6200-506 Covilhã, Portugal; 4Health Sciences Faculty, University of Beira Interior (FCS-UBI), 6200-506 Covilhã, Portugal

**Keywords:** immunological checkpoint inhibitors, atezolizumab, avelumab, cemiplimab, durvalumab, ipilimumab, nivolumab, pembrolizumab

## Abstract

By increasing the activity of the immune system, immune checkpoint inhibitors (ICPI) can have adverse inflammatory effects, which are referred to as immune-related adverse effects (irAEs). In this review, we present the recommendations for the appropriate identification and treatment of irAEs associated with ICPI to increase the safety and effectiveness of therapy with these immuno-oncological drugs. Several guidelines to manage irAEs adopted by different American and European societies in the field of oncology were identified. A narrative review of the several strategies adopted to manage irAEs was performed. With close clinical surveillance, ICPI can be used even in patients who have mild irAEs. Moderate to severe events require early detection and appropriate treatment, particularly in patients with a history of transplantation or pre-existing autoimmune disease. In most cases, adverse reactions can be treated with the interruption of treatment and/or supportive therapy, which includes, in serious adverse reactions, the administration of immunosuppressants. The identification and treatment of irAEs in the early stages may allow patients to resume therapy with ICPI. This review is an instrument to support healthcare professionals involved in the treatment and monitoring of patients who are administered ICPI, contributing to the timely identification and management of irAEs.

## 1. Introduction

Immunotherapy increases the individual’s own immune system’s ability to fight diseases, and in recent years, it has been a promising source of new treatments for oncological pathologies. Among the several immunotherapeutic strategies available, the inhibition of immune system checkpoint proteins has revealed significant beneficial effects in the treatment of several types of cancer [1].

The drugs called immunological checkpoint inhibitors (ICPI) have revolutionized the treatment of several oncological pathologies. These monoclonal antibodies target the signaling pathways of the immune system that involve the programmed cell death protein 1 (PD-1), programmed cell death ligand 1 (PD-L1), and cytotoxic T-lymphocyte antigen 4 (CTLA-4) molecules, contributing to restore the immune responses against neoplastic cells [2].

Despite the important clinical benefits of immunotherapy with the use of ICPI, including increased overall survival, these drugs are associated with significant adverse events related to the immune system, which can affect any organ or system of the body, the most frequently being gastrointestinal tract, endocrine glands, skin, and liver [1,2,3].

The aim of this work is to review the ICPI currently available in the pharmaceutical market and the main adverse effects associated with these drugs, as well as to review the recommendations for monitoring theses adverse effects, with a view to increasing the safety and effectiveness of pharmacological therapy with these immunomodulators used in cancer therapy.

## 2. Immunological Checkpoint Inhibitors Available in the Pharmaceutical Market and Main Associated Adverse Effects

ICPI lead to an increase in the anti-tumor activity of the immune system by blocking the intrinsic negative regulators of immunity, such as the CTLA-4 receptor (cytotoxic T-lymphocyte antigen 4) of cytotoxic T lymphocytes, the PD-1 receptor (programmed cell death protein 1) of cytotoxic T lymphocytes, or their PD-L1 ligand (programmed cell death ligand 1) in tumor cells [1,2]. Thus, ICPI inhibit the connection between the aforementioned cytotoxic T lymphocyte receptors and the corresponding cell ligands neoplastic cells, preventing the development of tolerance by cytotoxic T lymphocytes and, consequently, preventing neoplastic cells from escaping cell death mediated by those lymphocytes (Figure 1) [1,2]. It should be noted that under normal conditions (i.e., in the absence of oncological pathology or in tissues without oncological pathology), the aforementioned ligands are presented to cytotoxic T lymphocytes by antigen-presenting cells (APC), which allows the development of tolerance for the body’s own cells.

In several neoplastic pathologies, ICPI have led to an increase in the overall survival of cancer patients. Table 1 shows the various ICPI approved in July 2020 by the U.S. Food and Drug Administration (U.S. FDA) and the European Medicines Agency (EMA), as well as their mechanism of action.

Table 2 shows the authorized therapeutic indications of the ICPI available in the pharmaceutical market.

Cancer patients who receive ICPI may experience a specific set of adverse drug reactions, which are often referred to as immune-related adverse events (irAEs) [4,5]. These adverse reactions may involve several organs from different systems of the organism (Figure 2), and although they are generally mild in intensity, serious, irreversible, or even potentially fatal, adverse reactions can sometimes occur [1,2,6]. Since ICPI therapies are relatively recent, few doctors have clinical experience in recognizing and treating the adverse effects associated with these drugs.

The main adverse effects associated with ICPI are shown in Table 3, and more detailed information is presented in Appendix A. Since there are no prospective clinical studies that have defined strategies for the proper monitoring of adverse reactions related to the immune system, clinical practice in this area has been revealed as quite variable.

Several recommendations have been published to assist doctors and other health professionals to diagnose and identify adverse reactions related to the immune system associated with ICPI. These recommendations have also been aimed at harmonizing the monitoring of adverse reactions associated with ICPI. Examples of these recommendations are the guidelines of the European Society for Medical Oncology (ESMO) [3], the guidelines of the Society for Immunotherapy of Cancer (SITC) [7] and, more recently, the guidelines of the National Cancer Control Network (NCCN) [5], which were developed in collaboration with the American Society of Clinical Oncology (ASCO). More recently, in June 2018, ASCO published guidelines for clinical practice that address the adverse effects associated with ICPI—which are related to the immune system, based on the various systems or organs of the human body—together with recommendations for the diagnostic investigation and respective monitoring/treatment [4]. The development of these latest guidelines involved a systematic review of the literature and a consensus process, which was obtained from a multidisciplinary and multi-organizational panel of experts, consisting of doctors of various specialties, nurses, and specialists in clinical trials and advocacy [4].

The wide range of adverse drug reactions, involving the immune system, associated with ICPI, requires a synergistic multidisciplinary approach, involving several health professionals, for their monitoring, with the hospital pharmacist playing an important role in this multidisciplinary team, with a view to preventing and minimizing the adverse drug reactions mentioned.

It is also important to highlight that the complexity and multifactorial characteristics of neoplastic diseases require a molecular pathological epidemiology (MPE) approach [8]. The exposure to different risk factors influences specific carcinogenic mechanisms and different degrees of carcinogenicity. The integration of microbiology in the MPE model helps healthcare professionals and researchers understand the complex interactions existing between tumor cells, the immune system, and microorganisms in the tumor microenvironment during the carcinogenic process [9]. In order to optimize the effectiveness and safety of immunotherapies directed to immunological checkpoints, it is also essential to investigate and evaluate the influence of several immunomodulatory factors (such as environmental, dietary, lifestyle, microbial and pharmacological factors) in the response to immunotherapy.

## 3. Relevant Issues for the Monitoring and Treatment of Immune-Related Adverse Reactions

Postow et al. mentioned several issues that are relevant for different health professionals for the monitoring and treatment of adverse reactions that are related to the immune system, andassociated with ICPI [1]:

(1) Why do these adverse reactions occur?

The exact pathophysiology underlying the adverse events related to the immune system is unknown, but it is believed to be related to the role that immunological checkpoints play in maintaining immune homeostasis. For example, CTLA-4 acts by inhibiting an immune response in several ways, such as attenuating T cell activation in an early stage of the immune response. Translational studies in patients with adverse reactions related to the immune system have shown that in these adverse events, T cells, antibodies, and cytokines may be involved.

(2) How can these adverse reactions be treated?

As no prospective clinical study has defined the best treatment approaches, health professionals must base their work on consensus recommendations issued by experts. Most adverse events related to the immune system are treated effectively, delaying the administration of the drug or inducing temporary immunosuppression. Glucocorticoids (e.g., prednisolone) are generally the first-line immunosuppressive therapy. If glucocorticoids are initially ineffective, additional immunosuppressive agents (e.g., infliximab, adalimumab, golimumab, etanercept, mycophenolate, and mofetil) can be used.

(3) When do these adverse reactions occur?

The onset of these adverse reactions can vary from the first few weeks to months after starting treatment. They can potentially occur at any time, even after discontinuing treatment. Dermatological adverse reactions are usually the first to appear, as seen in clinical trials involving anti-CTLA-4 and anti-PD-1 therapies. It is still unknown whether the blocking of immunological checkpoints is associated with a risk of late toxicity. This issue becomes even more important for patients with neoplasms in early stages, in which life expectancy can be measured in decades.

(4) Why is it that only a few patients have these adverse reactions?

It is unknown why some patients experience adverse reactions related to the immune system. Research studies are currently underway to analyze whether genetic factors or the composition of the host microbiota are related to the risk of developing the aforementioned adverse reactions. Pre-clinical and clinical data suggest that certain bacterial species are linked to the effectiveness of blocking immunological checkpoints, which increases the possibility that variations in the patient’s gastrointestinal flora, which affect the host’s immunity, influence the risk of immunological adverse reactions.

(5) Are these adverse reactions in any way related to the effectiveness of the treatment?

The available data are contradictory, and it is still unclear whether the incidence of adverse reactions related to the immune system is associated with improved treatment effectiveness. The general consensus is that these reactions are not necessary for patients to benefit from treatment. However, some authors have noted that specific events, such as vitiligo, may be more clearly associated with treatment effectiveness.

(6) Does immunosuppression reduce the anti-tumor efficacy of ICPI therapy?

Retrospective studies have shown that clinical results are similar in patients who need immunosuppression to treat adverse events related to the immune system and in those who do not need such treatment. Beneficial responses may persist despite the use of immunosuppression to treat adverse events related to the immune system. However, additional studies are needed to analyze the potential relationship between the various aspects of immunosuppression (e.g., dosing schedule and duration of treatment) and clinical results in terms of anti-tumor effectiveness.

(7) Does immunosuppression have unintended adverse reactions?

Oncologists and other healthcare professionals need to be aware that immunosuppression carries additional risks. Specifically, the side effects of glucocorticoids can result in hyperglycemia, fluid retention, anxiety, and iatrogenic adrenal insufficiency. In addition, immunosuppression is a risk factor for subsequent opportunistic infections, such as Aspergillus fumigatus pneumonia and cytomegalovirus hepatitis.

(8) Is it safe to restart pharmacological therapy with ICPI after a serious adverse event?

Prospective data from clinical trials are limited as to the safety of resuming the blocking of immunological checkpoints after resolution of the adverse event. This is due to the fact that study protocols generally require that treatment be permanently discontinued if a patient experiences a serious adverse event related to the immune system.

Retrospective studies have shown that immune-related adverse events associated with the anti-CTLA-4 class may not necessarily occur again during treatment with another agent, such as a PD-1 inhibitor. The safety of restarting treatment depends on the severity of the previous event, the availability of alternative treatment options, and the stage of the cancer. However, restarting treatment should not be performed in case of life-threatening toxicity, particularly cardiac, pulmonary, or neurological toxicity.

(9) Is it necessary to restart treatment after the resolution of adverse reactions?

Regarding this issue, the data remain limited. Retrospective studies indicate that patients who had a favorable response to blocking immunological checkpoints and discontinued treatment due to adverse events related to the immune system generally maintain that favorable response. Prospective studies are needed to assess whether restarting immunotherapy is warranted.

(10) Is it safe to use ICPI in patients at risk of developing the aforementioned adverse reactions?

Patients at increased risk of adverse events related to the immune system, such as patients with pre-existing autoimmune disease, can still benefit from blocking immunological checkpoints. Age should not be a factor in excluding patients from treatment.

Subgroup analyses, from prospective and retrospective studies, suggest that the effectiveness of blocking immunological checkpoints in older adults is similar to the effectiveness seen in young adults, without an increase in adverse events related to the immune system [10].

## 4. Management of Adverse Effects Associated with Immunological Checkpoint Inhibitors

Below are some of the main recommendations for monitoring adverse reactions associated with ICPI, depending on the severity of each toxicity related to the immune system [11]. The definition of the degrees of toxicity mentioned below is in accordance with the Criteria Terminology for Adverse Events (version 5.0) [12].

Education: Oncologists and other health professionals should provide updated education/information to patients and their caregivers regarding immunotherapies, including education on potential adverse events related to the immune system, prior to the administration of ICPI, during treatment, and after the completion of treatment. Health professionals should also explain to patients and caregivers, using simple and accessible language, the mechanism of action of the specific treatment involved.

Assume an attitude of “presumption of guilt”: Oncologists and other health professionals should consider any new symptom (or new adverse reaction) in a patient who is receiving immunotherapy with a “high level of suspicion”, because such symptom may be related to the ICPI that the patient is receiving.

Grade 1 toxicities: In general, when grade 1 adverse events arise, patients should continue to use the ICPI, but should be monitored closely. Exceptions include adverse cardiac, neurological, and hematological events, which should lead to suspension of treatment.

Grade 2 toxicities: In general, when grade 2 adverse events arise, treatment should be stopped. Oncologists should consider restarting therapy when the patient’s symptoms and/or laboratory values are reduced to an adverse event of grade 1 or less. In the presence of grade 2 toxicities, patients should receive corticosteroids (i.e., prednisone or equivalent), with an initial dose of 0.5 to 1 mg/kg per day.

Restart therapy cautiously: It is advisable to exercise caution when restarting the administration of an ICPI after the patient’s symptoms or laboratory values return to ≤ 1 degree of toxicity. Physicians should be especially vigilant when attempting to restart treatment in patients who have had early-onset adverse events. Adjusting the dose of ICPI is not recommended.

Grade 3 toxicities: ICPI treatment should be discontinued for all grade 3 adverse events. These patients should receive high doses of corticosteroids that should be decreased over at least 4 to 6 weeks. Infliximab can be used for some adverse events if the patient’s symptoms do not improve within 48 to 72 hours after starting treatment with high doses of corticosteroids.

Grade 4 toxicities: The permanent discontinuation of ICPI is recommended when grade 4 toxicities occur, with the exception of endocrinopathies that can be controlled by hormone replacement (e.g., levothyroxine in hypothyroidism). In fact, endocrinopathies induced by ICPI are most often easily equilibrated by hormone replacement (in the case of deficiency) or improved by symptomatic treatment in the case of hyperfunction [13].

Table 4 describes the most relevant recommendations for the management/treatment of the main adverse reactions, related to the immune system, associated with ICPI. For more detailed information regarding the diagnosis and treatment of those adverse reactions, it is advisable to consult the recently published guidelines/consensus recommendations from The American Society of Clinical Oncology (ASCO) [4], National Comprehensive Cancer Network (NCCN) [5], Society for Immunotherapy of Cancer (SITC) [7], and European Society for Medical Oncology (ESMO) [3].

It should be noted that the incidence of adverse reactions related to the immune system varies with the prescribed ICPI and whether it is being administered as monotherapy or in combination (in the case of the association ipilimumab + nivolumab), depending also on the patient’s initial risk factors [2,14]. For example, CTLA-4 inhibitors are associated with higher rates of gastrointestinal effects and adrenal disorders, while PD-1/PD-L1 inhibitors are more associated with hypothyroidism and pneumonitis [2,14]. In addition, although the prescribed dose does not necessarily have an impact on the level of toxicity of PD-1/PD-L1 inhibitors, the same is not true with CTLA-4 inhibitors [14]. The period of time between the start of treatment and the onset of adverse reactions also differs between the various ICPI. With PD-1/PD-L1 inhibitors, symptoms are typically seen 4 to 12 weeks after the initiation of treatment, but with the combination of CTLA-4/PD-1 inhibition, adverse reactions generally arise much earlier [14].

In fact, the results of clinical trials indicate that most adverse events experienced by patients receiving ICPI are autoimmune in nature and generally occur during the first three months of therapy, although some may occur after the final dose has been administered [15]. Although toxicities associated with ICPI follow a predictable initial pattern (Box 1), toxicities associated with ipilimumab appear to be dose related, unlike what happens with nivolumab and pembrolizumab [3].

Box 1Initial toxicity pattern observed with ICPI.

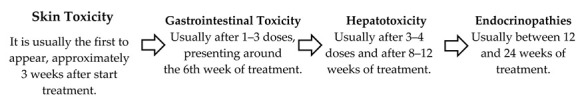



## 5. Treatment of Steroid-Refractory Immune-Related Adverse Effects

Steroids are the first-line therapy against most irAEs; however, when these adverse events are steroid-refractory, therapeutic options of immunosuppressant nature besides steroids are recommended [16,17].

In addition, giving high doses of steroids, especially for a long time period, can induce osteoporosis (which can be reduced by alendronate and calcium administration), infections (which can be avoided with the prophylactic administration of antibiotics and/or antifungals) and other adverse events such as hyperglycemia, muscle weakness, and edema [18,19,20]. If taken orally, they may predispose for oral candidiasis [21].

Early treatment with immunosuppressives or a combination of those with steroids may prevent further damage and allow faster irAEs resolution and lower steroids use [18]. Immunosuppressive drugs, including steroids, do not seem to negatively affect cancer response but may predispose for infection [11,22,23]. Therapeutic options of immunosuppressant nature besides steroids are recommended (Table 5). 

## 6. Role of the Hospital Pharmacist in Educating the Patient and Family or Caregivers

There is a need for hospital pharmacists to be very involved in the entire continuous process of treating patients who receive ICPI, and it is essential to take into account aspects such as the anticipation, evaluation, monitoring, and observation of toxicity [14]. Furthermore, in this entire continuous process of treatment, the education of doctors and patients/caregivers, by pharmacists, is a key factor.

Anticipation: Doctors often ask whether a patient is suitable for treatment with an ICPI and which specific inhibitor they should prescribe. It all comes down to a risk–benefit ratio in the specific patient concerned. Before starting a possible treatment with ICPI, doctors should take into consideration the patient’s clinical history, including autoimmune diseases or previous transplants, current pharmacotherapy, as well as obtaining laboratory tests or more specific assessments, based on the individual characteristics of the patient concerned.

Assessment: If a patient develops an adverse event related to the immune system, although the multidisciplinary health team must be on high alert (ICPI “presumed guilty” attitude), it is important not to assume immediately, without a more detailed assessment, that the ICPI is to blame. In many cases, it is necessary to consider the progression of the cancer disease itself, as well as other drugs prescribed concomitantly or the occurrence of infections. For example, if a patient has neurological toxicity of recent onset, the cause may be the progression of the disease (instead of the prescribed ICPI), especially in the case of metastatic melanoma.

Monitoring: The pharmacist must have knowledge of national and international guidelines/recommendations for the treatment of adverse reactions related to the immune system, with basic knowledge of the classification of adverse events being essential according to the “Adverse Event Terminology Criteria” [12]. In most cases of grade 3 or 4 immunological adverse events, treatment with ICPI should be discontinued and doses of corticosteroids increased. In this regard, it is important to take into account the need to slowly reduce corticosteroids, which is a reduction that should be made over at least 4 weeks. Patients who responded more slowly to corticosteroids (“steroid-refractory disease”) will need an even longer period of discontinuation, carried out for 6 to 12 weeks.

Observation: In the observation phase of the continuous treatment process, the multidisciplinary health team should focus on the time to resolve toxicity, on the continuous and gradual reduction of corticosteroids and on other drugs prescribed concurrently. For example, if the patient is taking non-steroidal anti-inflammatory drugs (NSAIDs), the possibility of discontinuing these drugs should be considered. If this is not possible, proton pump inhibitors should be considered to reduce the risk of gastrointestinal bleeding. When patients are in a process of reducing corticosteroids for a prolonged period, prophylaxis of infection by *Pneumocystis jirovecii* and antifungal prophylaxis should be considered in specific patients [14]. Alternatively, in the presence of certain adverse reactions related to the immune system, the prescription of ICPI may be temporarily suspended until those adverse reactions are resolved.

Several studies have revealed that there was no significant difference in clinical outcomes between patients who discontinued treatment with ICPI due to an adverse event related to the immune system and those who remained on treatment or received corticosteroids [35,36]. The pharmacist should have a role in the education of doctors, stressing the importance of stopping the inflammatory cascade early.

Education: In each of the previous steps, pharmacists will play a key role in educating doctors and patients/caregivers. Pharmacists should provide patients with basic immunotherapy concepts that help them understand the need for careful surveillance during and after therapy and the importance of immediately identifying adverse reactions. Patient education and pharmaceutical care can contribute to improving their quality of life, minimizing treatment delays or early discontinuation of therapy.

Many patients with previous experience in chemotherapy may have preconceived notions about what their new experiences of treatment with ICPI will be like. Patient education should include the transmission of information about immune activation and how responses to immunotherapy differ from those of chemotherapy. Specifically, immunotherapy may take longer to trigger a response when compared to conventional chemotherapy, and patients may experience stable cancer disease or even progression after initial treatment before favorable results are observed. In addition, side effects tend to be characterized by inflammation and require vigilance in observation and immediate notification to healthcare professionals to facilitate timely intervention. Patients need to be educated about the unique adverse reactions attributed to immunotherapy, which can be unexpected.

Table 6 is an example of a patient education tool, which can be dispensed by pharmacists to patients and family members, to convey critical information related to immunotherapy [2].

## 7. Conclusions

With the continuous development of new immuno-cancer drugs and their subsequent authorization and entry into the pharmaceutical market and the clinical care environment, an upward learning curve will continue to be observed. Due to their different mechanisms of action, ICPI cause not only distinct clinical responses, but also unique adverse events that differ from those observed with the most well-known cytotoxic agents.

While many of the adverse reactions related to the immune system are mild to moderate in nature, serious and life-threatening adverse reactions can also occur. Fortunately, by increasing the awareness of the multidisciplinary health team and the patient/caregivers and the immediate identification and early and appropriate treatment, many of these adverse events can be reversed. The identification and treatment of adverse effects in the early stages may allow patients to resume therapy with ICPI after resolving those adverse reactions related to the immune system.

With close clinical surveillance, ICPI can be used even in patients who experience mild immune system-related adverse events. Moderate to severe events require early detection and appropriate treatment, particularly in patients with a history of transplantation or pre-existing autoimmune disease. In most cases, adverse reactions can be treated with the interruption of treatment and/or supportive therapy, with the involvement and collaboration of the entire multidisciplinary health team and the patient and caregivers being important.

Additional clinical studies are needed to elucidate the mechanisms of action of adverse events related to the immune system to develop more accurate treatments for these events. The development of international registries will help to collect data, in real clinical situations, from patient populations underrepresented in clinical trials.

This review article is an instrument to support health professionals involved in the treatment and follow-up of cancer patients to whom ICPI are administered, contributing to the timely identification and management of adverse effects related to the immune system. This review will allow healthcare professionals to become familiar with strategies and best practices to quickly recognize and collaborate in minimizing and treating the set of toxicities unique to ICPI.

## Figures and Tables

**Figure 1 vaccines-08-00575-f001:**
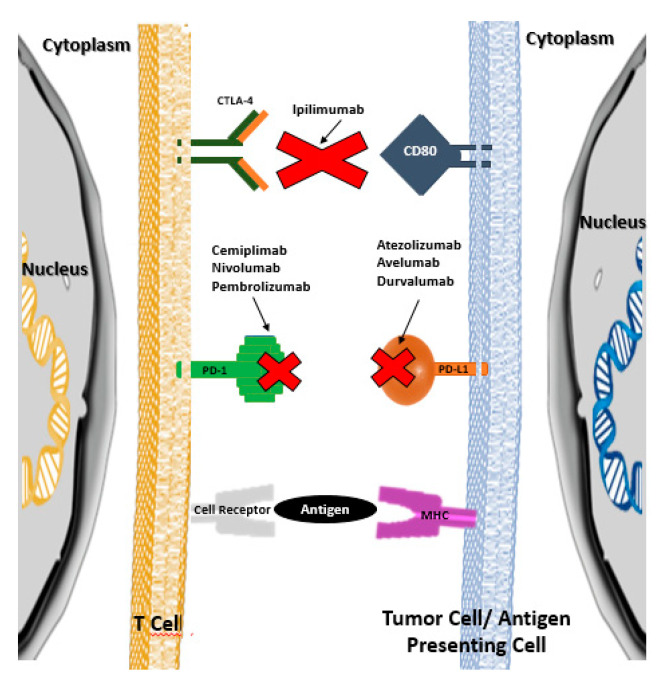
Therapeutic targets of immunological checkpoint inhibitors (adapted from references [1] and [2]).

**Figure 2 vaccines-08-00575-f002:**
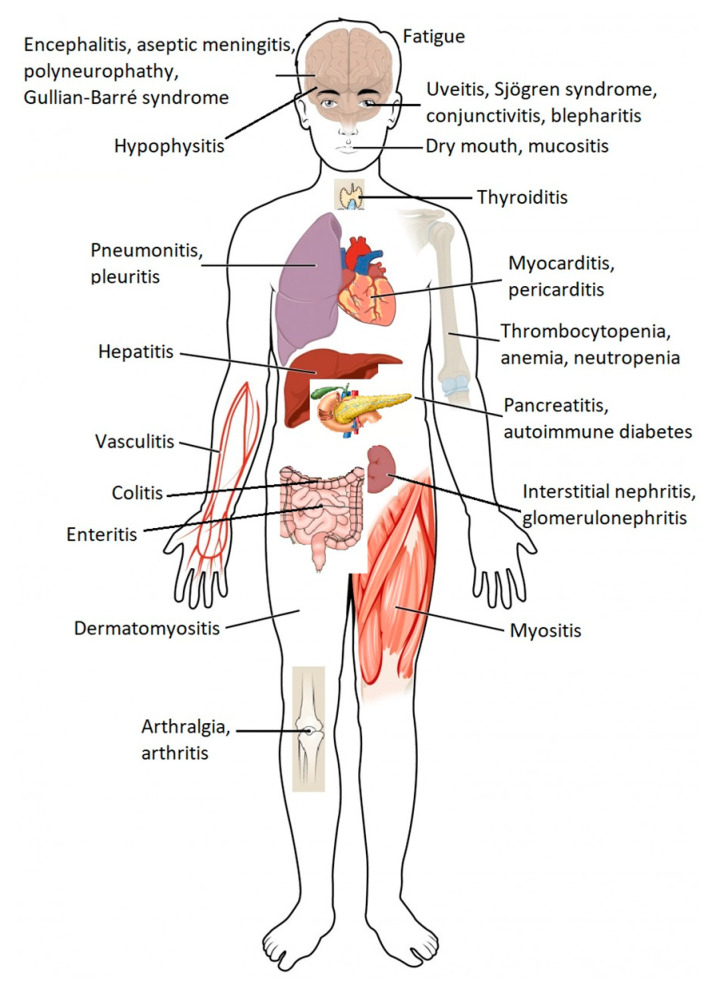
Common immune-related adverse events associated with immune checkpoint blockade by organs of the human body. Immune checkpoint blockade can result in the inflammation of any organ (adapted from ref. [1,6]).

**Table 1 vaccines-08-00575-t001:** Checkpoints inhibitors approved by the Food and Drug Administration (FDA) and the European Medicines Agency (EMA) presented according to the chronological order of introduction into the pharmaceutical market and their mechanism of action.

ICPI Monoclonal Antibody	Trade Name	Mechanism of Action
Ipilimumab	Yervoy	CTLA-4 inhibitor
Pembrolizumab	Keytruda	PD-1 inhibitor
Nivolumab	Opdivo	PD-1 inhibitor
Atezolizumab	Tecentriq	PD-L1 inhibitor
Avelumab	Bavencio	PD-L1 inhibitor
Durvalumab	Imfinzi	PD-L1 inhibitor
Cemiplimab	Libtayo	PD-1 inhibitor

CTLA-4: Cytotoxic T-lymphocyte antigen 4; ICPI: Immune checkpoint inhibitors; PD-1: Programmed cell death 1; PD-L1: Programmed cell death ligand 1.

**Table 2 vaccines-08-00575-t002:** The authorized therapeutic indications of the ICPI available in the pharmaceutical market.

	Ipilimu.	Pembrolizu.	Nivolu.	Atezolizu.	Avelu.	Durvalu.	Cemipli.
Breast cancer				✓			
Cervical cancer		✓					
Colorectal cancer	✓	✓	✓				
Cutaneous squamous cell carcinoma		✓					✓
Endometrial carcinoma		✓					
Esophageal cancer		✓	✓				
Gastric cancer		✓					
Head and neck cancer		✓	✓				
Hepatocellular carcinoma	✓	✓	✓	✓			
Hodgkin lymphoma		✓	✓				
Large B-cell lymphoma		✓					
Melanoma	✓	✓	✓	✓			
Merkel cell carcinoma		✓			✓		
Microsatellite instability-high tumors		✓					
Non-small cell lung cancer	✓	✓	✓	✓		✓	
Renal cell carcinoma	✓	✓	✓		✓		
Small cell lung cancer		✓	✓	✓		✓	
Tumor mutation burden-high tumors		✓					
Urothelial carcinoma		✓	✓	✓	✓	✓	

**Table 3 vaccines-08-00575-t003:** Most common adverse effects associated with ICPI (adapted from ref. [4]).

Affected Organ or System of the Human Body	Immune Related Adverse Effects
Skin	Bullous dermatosis Skin rash/Inflammatory dermatitis Severe skin reactions
Gastrointestinal	Colitis Hepatitis
Lung	Pneumonitis
Endocrine	Diabetes Hyperthyroidism (primary) Hypophysitis Primary adrenal insufficiency
Musculoskeletal	Inflammatory arthritis Myositis Polymyalgia rheumatica
Renal	Nephritis
Nervous System	Myasthenia gravis Guillain–Barré syndrome Peripheral neuropathy Autonomic neuropathy Aseptic meningitis Encephalitis Transverse myelitis
Hematological	Autoimmune hemolytic anemia Acquired thrombotic thrombocytopenic purpura Uremic hemolytic syndrome Aplastic anemia Lymphopenia Immune thrombocytopenia Acquired hemophilia
Cardiovascular	Myocarditis Pericarditis ArrhythmiasHeart failure associated with ventricular failure Vasculitis Venous thromboembolism
Ocular	Uveitis/Iritis Episcleritis Blepharitis

**Table 4 vaccines-08-00575-t004:** Recommendations for the treatment of the main adverse reactions, related to the immune system, associated with ICPI (adapted from ref. [2]).

Common Adverse Reactions	Research of Alternative/Non-Inflammatory Etiologies	Degree of Toxicity	Recommended Management of Immune-Related Adverse Events (irAEs)
Gastrointestinal Diarrhea/Colitis	Exclude infectious etiology (Clostridium difficile)	Grade 1 (Mild)	Symptomatic treatment Consider budesonide 9 mg/day Continue immunotherapy
		Grade 2 (Moderate)	Delay immunotherapy Methylprednisolone IV 0.5–1 mg/kg/day (or oral equivalent) Consider gastroenterology and colonoscopy consultation When improving to ≤ grade 1, reduce the dose for at least 4 weeks
		Grade 3–4 (Severe)	Stop immunotherapy Methylprednisolone IV 1–2 mg/kg/day When improving to ≤ grade 1, reduce the dose for at least 4 weeks If no improvement in symptoms within 48–72 h, consider 2nd line immunosuppression (infliximab)
Hepatitis	Evaluate for: - Alcohol intake - Concomitant drugs with hepatotoxic potential - Exclude biliary disease/biliary obstruction	Grade 1 (Mild)	Continue immunotherapy Repeat LFTs within 1 week
		Grade 2 (Moderate)	Delay immunotherapy Repeat LFTs every 3–5 days Methylprednisolone IV 0.5–1 mg/kg/day (or oral equivalent) When improving to mild or baseline, reduce the dose of steroids for at least 4 weeks
		Grade 3–4 (Severe)	Stop immunotherapy Increase the frequency of LFTs to 1–2 days Methylprednisolone IV 1–2 mg/kg/day Gastroenterology consultation If no improvement in symptoms within 48-72 h, consider 2nd line immunosuppression (infliximab)
Pneumonitis	Evaluate for: - Pulmonary embolism - Cardiac causes - Infectious etiology - COPD - Seasonal allergies/cough post-nasal drip	Grade 1 (Mild)	Delay immunotherapy Monitor symptoms Repeat chest X-ray in 2–4 weeks
		Grade 2 (Moderate)	Delay immunotherapy Monitor symptoms closely, consider hospitalization Re-image every 1–3 days Pneumology and infectious disease consultations, consider bronchoscopy Methylprednisolone IV 1–2 mg/kg/day (or oral equivalent) When symptoms improve, reduce the dose of steroids for at least 4 weeks
		Grade 3–4 (Severe)	Stop immunotherapy Methylprednisolone IV 2–4 mg/kg/day, discontinue steroids for a period of at least 6 weeks If no improvement in symptoms within 48–72 h, consider 2nd line immunosuppression (infliximab, mycophenolate mofetil, IVIG)
Dermatological adverse reactions	Exclude non-inflammatory causes (allergic reaction to other drugs, photosensitivity, etc.)	Grade 1 (Mild)	Continue immunotherapy Supportive therapy emollients, low-potency topical steroids, antihistamines
		Grade 2 (Moderate)	Continue immunotherapy Topical steroids of moderate-high potency If persistent, despite optimized topical treatment, consider methylprednisolone 0.5–1 mg/kg/day (or oral equivalent) If it improves slightly or resolves, reduce the dose of steroids for at least 4 weeks Consider dermatological evaluation and skin biopsy
		Grade 3–4 (Severe)	Delay immunotherapy Methylprednisolone IV 1–2 mg/kg/day (or oral equivalent) If it improves to mild or resolves, reduce the dose of steroids for at least 4 weeksConsider skin biopsy
Endocrinopathies	Exclude non-inflammatory etiology of symptoms	Grade 1 (Mild)	Continue immunotherapy If TSH is abnormal, add free T4 and T3 Consider morning cortisol and ACTH
		Grade 2 (Moderate)	TSH, free T4, morning cortisol and ACTHConsider pituitary MRIMethylprednisolone IV 1–2 mg/kg/day (or oral equivalent)If it improves, reduce the dose of steroids for at least 4 weeksHormone replacement therapy if indicatedEndocrinology consultation
		Grade 3–4 (Severe)	Delay or discontinue immunotherapy If adrenal crisis is suspected, exclude infection/sepsis, BP support Stress doses of mineralocorticosteroid

ACTH: Adrenocorticotrophin; COPD: Chronic obstructive pulmonary disease; GI: Gastrointestinal; IVIG: Intravenous immunoglobulin; IV: Intravenous; MRI: Magnetic resonance image; BP: Blood pressure; T3: Triiodothyronine; T4: Thyroxine; LFTs: Liver function tests; TSH: Thyroid stimulating hormone.

**Table 5 vaccines-08-00575-t005:** Therapeutic options of immunosuppressant nature besides steroids for irAEs treatment.

Immunosuppressive Drug/Commentary	Immune-Related Adverse Events (irAEs) Treated
Anti-IL6: tocilizumab. It is not expected that anti-tumoral response will cease, since concomitant blockade of PD-1/PD-L1 and IL-6 have a synergic anti-tumoral effect [17,24,25].	Severe or refractory arthritis, large vessel vasculitis, uveitis, myocarditis, myastenia gravis, pneumonitis, hepatitis, hypophisitis, colitis, pancreatitis, mediated coagulopathy.
Sulfasalazine [26].	Refractory rheumatologic irAEs.
Mycophenolate mofetil [17,27,28,29,30].	Hepatitis, pneumonitis, myocarditis.
Calcineurin inhibitors: tacrolimus (dose based on blood results) or cyclosporine [17,27,29,31,32].	Tacrolimus: hepatitis, myocarditis. Cyclosporine: enterocolitis, hepatitis.
Anti-TNFα blockade: infliximab, adalimumab, golimumab, etanercept, certolizumab [18,19,29,30,33,34].	May be a good option especially in cases of steroid refractory colitis, myocarditis, bile duct obstruction and pneumonitis, arthritis, nephritis, and uveitis.
IV immunoglobulins [16].	Guillain–Barré syndrome, subacute and chronic inflammatory demyelinating polineuropathy, thromnocytopenia, enteric neuropathy, ocular myositis, encephalitis, facial nerve palsy, myasthenia gravis, transverse myelitis.
Plasmapheresis [16].	Myastenia gravis, acute inflammatory demyelinating poliradiculoneuropathy, encephalitis.
Azathioprine: before administration, a thiopurine S-methyltransferase (TPMT) genotype or enzyme function test should be carried, because patients with lower TPMT activity have an increased risk of manifesting life-threatening bone marrow suppression [16,27].	Hepatitis.
Anti-CD20 depletion: rituximab, ofatumumab, obinutuzumab, ocrelizumab [16].	Systemic lupus erythematosus, antineutrophil cytoplasmic antibody-associated vasculitis, severe Sjögren’s syndrome, cutaneous vasculitis, nephritis, autoimmune autonomic ganglionopathy, sensory ganglionopathy, myasthenia gravis, transverse myelitis, enteric neuropathy, encephalitis, aseptic meningitis, hepatitis.
Anti-IL-17-blockade: ixekizumab, brodalumab, secukinumab [16].	Colitis, severe psoriasis refractory to anti-TNFα therapy, rheumatoid arthritis, anti-IL6 refractory irAEs.
Anti-IL-23 and anti-IL-12 blockade: ustekinumab [16].	Psoriasis and psoriatic arthritis.
Janus kinase inhibitor: tofacitinib [16].	Rheumatoid arthritis and ulcerative colitis.
Anti-B-cell strategy: belimumab [16].	Lupus erythematosus.
Cyclophosphamide [16].	Symptomatic sarcoidosis, pneumonitis, Guillian–Barré syndrome, Stevens–Johnson syndrome with central and neurologic symptoms, autoimmune autonomic ganglionopathy, sensory ganglionopathy, polyneuropathy, central neuritis.

**Table 6 vaccines-08-00575-t006:** Example of an education tool in the field of immunotherapy to be dispensed by pharmacists to patients and family members (adapted from ref. [2]).

Immunotherapy Alert and Education Card for Patients
What is immunotherapy? Cancer immunotherapy is a type of drug therapy that stimulates the body’s natural defenses to fight cancer.
How will immunotherapy be administered? Immunotherapy is administered intravenously (IV). It can be administered via peripheral or central venous (IV) access.
What adverse effects may arise during treatment? The onset of adverse effects may vary from patient to patient and with the prescribed therapy. Adverse effects can occur from 1–3 weeks after the start of treatment to months after the end of therapy. It is recommended to discuss any new adverse effects immediately with the oncologist, nurse, or pharmacist. The general adverse effects to be observed include: (1) Dermatological problems: rash, itching, dry skin, rash, changes in color; (2) Gastrointestinal problems: flatulence, abdominal bloating, nausea or vomiting, changes in the usual pattern of bowel movements, black, sticky, tar-like stools, or blood or mucus stools; (3) Liver problems: nausea or vomiting, loss of appetite, pain in the right side of the stomach, yellowing of the skin or whites of the eyes, dark urine, bleeding or bruising more often than normal; (4) Endocrine problems (especially thyroid, pituitary, and adrenal): fatigue, rapid heartbeat, weight loss or gain, increased sweating, hair loss, feeling cold, constipation, lower voice, muscle pain, dizziness or fainting, headaches that are persistent or headache that is unusual; (5) Lung problems: shortness of breath, chest pain, cough.
Who can I contact if I develop adverse effects? Name and contact of the oncologist: Contact of the hospital where the treatment is being carried out: Emergency phone: Nursing team contact: Hospital pharmacy contact:
How will adverse effects be treated? Adverse effects are treated based on the nature and severity of the symptoms. However, in general, most adverse effects may require the administration of a corticosteroid for treatment. The dosage and duration of treatment may vary depending on the intensity and severity of the symptoms and will be discussed with the attending physician.

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
