# Peer review of "Management of the Adverse Effects of Immune Checkpoint Inhibitors"

_vaccines, 2020, doi:10.3390/vaccines8040575_

Round 1

Reviewer 1 Report

Morgado et al revised the literature to describe the adverse effects of ICPI. The review is well written and organized. Furthermore some some additions are required.

Could authors provide a table reporting the percentage/the frequency with which each side effect manifests itself in relation to each ICPI. 

Why authors reported only ICPI for PD1, PDL1, CTLA and not for others ICP? Please, give explanation.

Please, provide figures with higher resolution.

Author Response

Morgado et al revised the literature to describe the adverse effects of ICPI. The review is well written and organized. Furthermore, some additions are required.

Could authors provide a table reporting the percentage/the frequency with which each side effect manifests itself in relation to each ICPI. 

R: Many thanks for your comments and for the suggestion. A table with more detailed information and frequency of each adverse effect was added as supplementary material (Supplementary Material S1). The information was collected in the Summary of Product Characteristics (SmPC) of each product.

Why authors reported only ICPI for PD1, PDL1, CTLA and not for others ICP? Please, give explanation.

R: The aim of this study was to review the ICPI currently available in the pharmaceutical market, and to our knowledge and according to FDA and EMA sites there are only ICPI drugs approved for target CTLA-4, PD-1, and PD-L1. In further studies, we will explore scientific bibliography about new potential therapeutic targets.

Please, provide figures with higher resolution.

R: Figures resolution was improved.

Reviewer 2 Report

This is a generally well-written review on adverse reactions to immunotherapy. I believe this can be a very useful article for future of oncology.

There are many environmental, dietary, and lifestyle factors that influence the microbiome (in both gut lumen and tissue), immune system, pathogenic mechanisms, and response to immunotherapy (including adverse reactions). The authors should discuss these points; influence of those factors, eg, smoking, alcohol, diet, obesity, microbiome, etc. on immunity. Personal differences in immune response likely result from these factors as well as our genetic differences. But very few studies assessed these factors. If we know which diet can reduce adverse effect, we can recommend such diet, but we have no data yet.  

In these contexts, as a future direction, research on dietary / lifestyle factors, microbiome, immunity, and personalized molecular biomarkers is needed. The authors should discuss molecular pathological epidemiology (MPE), which can investigate those factors in relation to molecular pathologies, immunity, and clinical outcomes including adverse reactions. MPE has been discussed in J Pathol 2019, Annu Rev Pathol 2019, etc. MPE research can be a promising direction and improve prediction of response and adverse reaction to immunotherapy or other forms of immune-based intervention. 

Author Response

Revewer 2

This is a generally well-written review on adverse reactions to immunotherapy. I believe this can be a very useful article for future of oncology.

 There are many environmental, dietary, and lifestyle factors that influence the microbiome (in both gut lumen and tissue), immune system, pathogenic mechanisms, and response to immunotherapy (including adverse reactions). The authors should discuss these points; influence of those factors, eg, smoking, alcohol, diet, obesity, microbiome, etc. on immunity. Personal differences in immune response likely result from these factors as well as our genetic differences. But very few studies assessed these factors. If we know which diet can reduce adverse effect, we can recommend such diet, but we have no data yet.  

 In these contexts, as a future direction, research on dietary / lifestyle factors, microbiome, immunity, and personalized molecular biomarkers is needed. The authors should discuss molecular pathological epidemiology (MPE), which can investigate those factors in relation to molecular pathologies, immunity, and clinical outcomes including adverse reactions. MPE has been discussed in J Pathol 2019, Annu Rev Pathol 2019, etc. MPE research can be a promising direction and improve prediction of response and adverse reaction to immunotherapy or other forms of immune-based intervention. 

R: Authors are thankful for the comments and for the suggestions. The following information was added to the manuscript. 

“It is also important to highlight that the complexity and multifactorial characteristics of neoplastic diseases requires a molecular pathological epidemiology (MPE) approach [8]. The exposure to different risk factors influences specific carcinogenic mechanisms and different degrees of carcinogenicity. The integration of microbiology in the MPE model helps healthcare professionals and researchers to understand the complex interactions existing between tumour cells, the immune system and microorganisms in the tumour microenvironment during the carcinogenic process [9]. In order to optimize the effectiveness and safety of immunotherapies directed to immunological checkpoints, it is also essential to investigate and evaluate the influence of several immunomodulatory factors (such as environmental, dietary, lifestyle, microbial and pharmacological factors) in the response to immunotherapy.”

And the following references were added to the references list:

[8] Ogino, S.; Nowak, J.A.; Hamada, T.; Milner, D.A., Jr.; Nishihara, R. Insights into Pathogenic Interactions Among Environment, Host, and Tumor at the Crossroads of Molecular Pathology and Epidemiology. Annu Rev Pathol 2019, 14, 83-103, doi:10.1146/annurev-pathmechdis-012418-012818.

[9] Hamada, T.; Nowak, J.A.; Milner, D.A., Jr.; Song, M.; Ogino, S. Integration of microbiology, molecular pathology, and epidemiology: a new paradigm to explore the pathogenesis of microbiome-driven neoplasms. J Pathol 2019, 247, 615-628, doi:10.1002/path.5236.“

Round 2

Reviewer 1 Report

Authors replied to all comments and the review is acceptable in the present form.